# Natural language processing augments comorbidity documentation in neurosurgical inpatient admissions

Rahul A. Sastry[1]*, Aayush Setty[1,2], David D. Liu[1], Bryan Zheng[1], Rohaid Ali[1], Robert J. Weil[3], G. Dean Roye[4], Curtis E. Doberstein[1], Adetokunbo A. Oyelese[1], Tianyi Niu[1], Ziya L. Gokaslan[1], Albert E. Telfeian[1]

1 Department of Neurosurgery, Warren Alpert Medical School, Rhode Island Hospital, Brown University, Providence, RI, United States of America, 2 Department of Computer Science, Brown University, Providence, RI, United States of America, 3 Department of Neurosurgery, Brain & Spine, Southcoast Health, Dartmouth, MA, United States of America, 4 Department of Surgery, Warren Alpert Medical School, Rhode Island Hospital, Brown University, Providence, RI, United States of America

☯ These authors contributed equally to this work.
* rahul.sastry@gmail.com

**Data Availability Statement:** All relevant anonymized data and code are within the manuscript and its Supporting Information files.

## Abstract

### Objective

To establish whether or not a natural language processing technique could identify two common inpatient neurosurgical comorbidities using only text reports of inpatient head imaging.

### Materials and methods

A training and testing dataset of reports of 979 CT or MRI scans of the brain for patients admitted to the neurosurgery service of a single hospital in June 2021 or to the Emergency Department between July 1–8, 2021, was identified. A variety of machine learning and deep learning algorithms utilizing natural language processing were trained on the training set (84% of the total cohort) and tested on the remaining images. A subset comparison cohort (n = 76) was then assessed to compare output of the best algorithm against real-life inpatient documentation.

### Results

For "brain compression", a random forest classifier outperformed other candidate algorithms with an accuracy of 0.81 and area under the curve of 0.90 in the testing dataset. For "brain edema", a random forest classifier again outperformed other candidate algorithms with an accuracy of 0.92 and AUC of 0.94 in the testing dataset. In the provider comparison dataset, for "brain compression," the random forest algorithm demonstrated better accuracy (0.76 vs 0.70) and sensitivity (0.73 vs 0.43) than provider documentation. For "brain edema," the algorithm again demonstrated better accuracy (0.92 vs 0.84) and AUC (0.45 vs 0.09) than provider documentation.

**Funding:** The author(s) received no specific funding for this work.

## Discussion

A natural language processing-based machine learning algorithm can reliably and reproducibly identify selected common neurosurgical comorbidities from radiology reports.

## Conclusion

This result may justify the use of machine learning-based decision support to augment provider documentation.

## Introduction

Timely and accurate medical documentation is a quality and safety imperative. Precise documentation can advance efficacious inpatient care, enhance transitions across the healthcare ecosystem, reduce needless variation and excess utilization, facilitate clinical research efforts, and capture the intensity and quality of care, on which hospital reimbursements are based [1–3]. Education and training regarding best practices in documentation, however, can be perceived as extraneous or of minimal importance, especially in the context of resident education and training. Such inattention can result in substantial underestimations of the intensity of care provided to operative and non-operative surgical inpatients [2–6]. Estimated revenue losses of up to 40% have clear and obvious consequences for hospital operations, particularly in the context of hospitalized trauma patients who do not undergo surgical intervention [3]. An array of interventions, including provider education, constant clinician review of electronic medical records (EMR), manually generated documentation queries, and others, have been implemented at various centers. However, they are often additive to the work of busy clinicians and trainees, who already spend historically large amounts of time on documentation, or by expanding the numbers of clinical documentation staff to constantly assess provider practices [5, 7–10]. These additive measures are of the work harder, not smarter, framework and have been documented across healthcare to be a growing source of clinician discontent and burnout [11, 12].

In the United States, inpatient reimbursements are determined by broad classifications of patient diagnoses known as diagnosis-related groups (DRGs), which were originally implemented as part of Medicare's Prospective Payment System in 1983 [1, 13, 14]. Medicare Severity DRGs (MS-DRGs), the most common system used in the United States, are stratified into three categories: (1) DRG without complication or comorbidity (CC) and without major CC; (2) DRG with a CC; and (3) DRG with MCC [1, 14]. Inpatient and relevant outpatient documentation of pertinent medical and surgical diagnoses, as well as the specific treatments or interventions that treat these diagnoses, determines the CCs and MCCs used as secondary diagnoses during and after admission. In this context, our neurosurgery department at a large American level 1 trauma center recently implemented a provider-based initiative to improve inpatient documentation and comorbidity capture rates [4].

Given the success of machine learning (ML) approaches in a variety of medical contexts [15–17], we hypothesized that a natural language processing (NLP)-based ML algorithm may be able to identify neurosurgical inpatients likely to have 1 or more commonly encountered CC/MCCs based solely on text interpretations of computed tomography (CT) or magnetic resonance imaging (MRI) reports obtained during hospital admission regardless of underlying pathology.

## Materials and methods

### Patient cohort

The protocol for this study was reviewed and approved by the Institutional Review Board of Rhode Island Hospital (Providence, RI). As the proposed research was a retrospective observational study, the need for patient consent was waived by the aforementioned Institutional Review Board. Data were fully anonymized at the time of chart review. A retrospective cohort of 979 images was comprised of all scans and respective radiological impressions of patients admitted to the neurosurgery service in June 2021 who underwent either CT or MRI of the brain and all scans of patients seen in the emergency department from July 1–8, 2021 who underwent either CT or MRI of the brain. This cohort was devised in order to include an appropriate number of positive and negative controls for algorithm training and development even though the target population of this effort consists exclusively of neurosurgical admissions. Given the nature of these inclusion criteria, in some cases, multiple scans were included for a given patient/admission. A separate provider comparison cohort, which was comprised of 76 patients who were admitted to the neurosurgery service in October 2021 and underwent inpatient CT or MRI of the brain, was also identified in order to facilitate comparison between algorithm performance and real-world documentation. October 2021 was selected as a representative month because it reflected steady state documentation practices after the recent implementation of a documentation improvement protocol and progress note template [4]. In this cohort, only the first scan obtained within our hospital's system, regardless of indication or modality, was included (thus resulting in one scan per patient). This cohort was specifically only used as a subset of the test data so that we can compare model performance to provider performance on the entire cohort. The combined dataset (n = 1055) was split into a training cohort (n = 885, 83.9%) and a testing set (n = 170, 16.2%) for the purpose of algorithm selection and training. Class imbalance was a major consideration in our data split structure as model fitting can be biased by highly imbalanced datasets. We ensured that our training and testing set had relatively similar class proportions **Table 1**. Images were not excluded on the basis of elective vs. emergent admission or surgical vs. non-surgical pathologies.

### Gold labels

All 1,055 patient images were reviewed by a single author (RAS) in a blinded fashion and were assessed for the presence or absence of "brain compression" or "brain edema," both of which are common neurosurgical CC/MCCs that were the primary targets of a recent intradepartmental documentation improvement effort [4].

### Human prediction

Records for patients in the provider comparison cohort, the temporal range of which was chosen to reflect documentation practices *after* successful implementation of a provider-education intervention in late 2020, were also manually queried for discharge summary documentation of "brain compression" and "brain edema"; as such, for patients in this cohort, presence or

**Table 1. Class proportions for brain compression and edema.**

| Dataset | Compression Positive | Compression Negative | Edema Positive | Edema Negative |
|---|---|---|---|---|
| Testing | 23.16% | 76.84% | 11.41% | 88.59% |
| Training | 28.82% | 71.18% | 14.71% | 85.29% |

absence of either term in the discharge summary were used to assess the performance of real-world provider documentation against the gold standard of author review.

## Data pre-processing

We only used the impression texts of CT and MRI radiology reports to predict "brain compression" and "brain edema" classifications. All word/data tokenization was completed using the Natural Language Toolkit (NLTK) [18] package in the Python programming language (Python Software Foundation, https://www.python.org/). All texts were first "tokenized" into single word vectors by splitting the text on white space thereby one word becoming one word "token". The list of tokens was then parsed and all words were casted to lowercase, all stop words were removed, and all punctuation was removed to isolate significant word tokens. The list of word vectors was then scored based on two different word tokenizing strategies: term frequency-inverse document frequency (TF-IDF) and frequency (TF) (**Fig 1**). These word vectors were then fed into ML and deep learning (DL) algorithms with a bag of words technique to predict lesion classification. Bag of words featurization allows for sentences to be vectorized based on the words they contain. The dimension of the sentence vector space is set to the number of unique word tokens where each index of the vector is representative of a unique word. Each sentence vector is constructed by either using a TF approach or a TF-IDF approach. The TF approach assigns values to each index in a sentence vector based on the frequency of that word occurring in the sentence. The TF-IDF approach discounts words that occur in high frequency in the corpus by a discounting factor of $log(N/df)$ where N represents the number of

*Impression 1: "Acute subdural hematoma centered in the right frontal temporal region."*

*Impression 2: "Small acute subdural hemorrhages localized in the left frontal region."*

*N = 2 (number of documents in the corpus)*

| acute | in | hematoma | centered | small | the | frontal | region | temporal | subdural | left | right | hemorrhages | localized |
|---|---|---|---|---|---|---|---|---|---|---|---|---|---|
| df – Number of documents with the term present | | | | | | | | | | | | | |
| 2 | 2 | 1 | 1 | 1 | 2 | 2 | 2 | 1 | 2 | 1 | 1 | 1 | 1 |
| IDF Value = log(N/df) | | | | | | | | | | | | | |
| 0 | 0 | 0.3 | 0.3 | 0.3 | 0 | 0 | 0 | 0.3 | 0 | 0.3 | 0.3 | 0.3 | 0.3 |

Impression 1 Tokenization

| acute | in | hematoma | centered | small | the | frontal | region | temporal | subdural | left | right | hemorrhages | localized |
|---|---|---|---|---|---|---|---|---|---|---|---|---|---|
| Frequency Tokenization (TF) | | | | | | | | | | | | | |
| 0.1 | 0.1 | 0.1 | 0.1 | 0.0 | 0.1 | 0.1 | 0.1 | 0.1 | 0.1 | 0.0 | 0.1 | 0.0 | 0.0 |
| TF-IDF Tokenization – TF*IDF | | | | | | | | | | | | | |
| 0.0 | 0.0 | 0.03 | 0.03 | 0.0 | 0.0 | 0.0 | 0.0 | 0.03 | 0.0 | 0.0 | 0.03 | 0.0 | 0.0 |

Impression 2 Tokenization

| acute | in | hematoma | centered | small | the | frontal | region | temporal | subdural | left | right | hemorrhages | localized |
|---|---|---|---|---|---|---|---|---|---|---|---|---|---|
| Frequency Tokenization | | | | | | | | | | | | | |
| 0.1 | 0.1 | 0.0 | 0.0 | 0.1 | 0.1 | 0.1 | 0.1 | 0.0 | 0.1 | 0.1 | 0.0 | 0.1 | 0.1 |
| TF-IDF Tokenization | | | | | | | | | | | | | |
| 0.0 | 0.0 | 0.0 | 0.0 | 0.03 | 0.0 | 0.0 | 0.0 | 0.0 | 0.0 | 0.03 | 0.0 | 0.03 | 0.03 |

**Fig 1. Example of frequency and TF-IDF tokenization strategies illustrating how TF-IDF controls for words that frequently occur in the corpus.**
TF-IDF = Term Frequency Inverse Document Frequency.

reports present in the dataset and df representing the number of documents that a specific term was present in. The values at each index of a sentence vector constructed by using the TF-IDF approach is the product of the discounting factor and the respective TF value (Fig 1).

Overall, the entire dataset of radiology reports only mentioned "compression" in 1.7% of the reports and mentioned "edema" in 15.5% of the reports. The presence of these specific tokens do not necessarily correlate deterministically with a positive label (27% and 54% of reports with compression and edema tokens were positive for compression and edema respectively) which demonstrates the need of a more sophisticated NLP approach for classification.

As previously noted, the primary patient cohort data was split into an 84% training set and 16% testing set. After tokenizing and preprocessing the radiology data, we used a multitude of ML and DL supervised learning models to predict lesion classifications.

### Machine learning and deep learning prediction

We used python packages scikit-learn Python [19] and Tensorflow [20] to fit various ML and DL models respectively. We trained a random forest, logistic, support vector machine, and Naive Bayes classifiers on both word tokenization techniques (TF-IDF and TF). As for DL models, we fit a single layer perceptron and a multilayer perceptron for classification. Each model was fit once for brain compression and once for brain edema; all models were binary classifiers. Both DL methods used a binary cross-entropy loss function, the random forest used a Gini Impurity loss function, and the rest of the ML classifiers used their respective default loss functions/techniques that were prebuilt in scikit-learn's library.

Hyperparameter optimization was carried out through grid search optimizing for area under the curve (AUC) to avoid biased model fitting due to our slightly unbalanced datasets. Each model was fit using 5-fold cross validation where the training data was split into 5 equally sized groups, and the model was trained with 4 out of the 5 folds where the last fold was used as a validation set. 5-fold validation was carried out with shuffle split to ensure training data was shuffled and randomized before being divided into folds. The training and validation accuracies were only used to select the best hyperparameters for each model and are not reported here. For each model, AUC and accuracy, the proportion of correct binary classifications among the samples in the test set, are reported in addition to other select metrics.

### Data availability

A de-identified dataset that supports the findings of this study is available on request. The data are not publicly available due to the use of protected health information.

### Results

Characteristics of the patient cohort and included imaging studies are summarized in **Table 2.** The performance of included ML and DL algorithms for prediction of "brain compression" among the testing cohort are presented in **Fig 2.** Among ML methods, a random forest classifier with TF-IDF tokenization outperformed other candidate algorithms with an accuracy of 0.81 (Standard Deviation [SD] 0.01) and AUC of 0.90 (SD 0.01). Among DL methods, a multilayer perceptron method with frequency tokenization outperformed other candidate algorithms with an accuracy of 0.78 (SD 0.02) and AUC of 0.88 (SD 0.01). The performance of included ML and DL algorithms for prediction of "brain edema" are presented in **Fig 3.** Among ML methods, the random forest classifier with term frequency-inverse document frequency again outperformed other candidate algorithms with an accuracy of 0.92 (SD 0.02) and AUC of 0.94 (SD 0.01). Among DL methods, a multi-layer perceptron method with term

**Table 2. Demographic and diagnostic characteristics for provider comparison cohort.**

| Patient Characteristics | Provider Comparison cohort (n = 76) |
|---|---|
| **Demographic Details** | |
| Male | 38 (50%) |
| Age (years) | 67 (± 15.1) |
| **Diagnostic Details** | |
| Indication for imaging | |
| Trauma | 37 (48.7%) |
| Vascular | 16 (20.5%) |
| Tumor | 9 (11.5%) |
| Postoperative | 6 (7.7%) |
| Shunt | 5 (6.4%) |
| Altered mental status | 2 (2.5%) |
| Infectious | 1 (1.2%) |
| Type of imaging | |
| CT without contrast | 66 (86.8%) |
| CT with contrast | 10 (13.1%) |
| MRI with contrast | 10 (13.1%) |

frequency-inverse document frequency outperformed other candidate algorithms with an accuracy of 0.89 (SD 0.01) and AUC of 0.87 (SD <0.01).

Receiver operating characteristic (ROC) curves for both random forest classifiers are shown in **Fig 4.** For the optimal compression classifier, we chose a point on the ROC curve that corresponded to a classifier with an accuracy of 0.81, specificity of 0.88, and sensitivity of 0.65. For the optimal edema classifier, we chose a point on the ROC curve that corresponded to a classifier with an accuracy of 0.92, specificity of 1.0, and sensitivity of 0.48. A more complete characterization of each ML and DL classifier's performance is broken down in **Table 3A and 3B**.

Based on these data, the random forest classifier was selected as the best performing algorithm and compared against discharge summary documentation in **Fig 5.** For documentation of "brain compression", the Random Forest ML algorithm demonstrated better accuracy (0.76 vs 0.70) and sensitivity (0.73 vs 0.43) than provider documentation. The logarithmic regression also performed very well with an identical accuracy (0.76) and a slightly higher sensitivity (0.76 vs 0.73) albeit having a slightly lower AUC when compared to the Random Forest (0.87 vs 0.89). For "brain edema," the Random Forest ML algorithm again demonstrated better accuracy (0.92 vs 0.84) and sensitivity (0.45 vs 0.09) than provider documentation. The logarithmic regression also performed very well with an identical accuracy (0.92) and a higher sensitivity (0.54 vs 0.45) albeit having a lower AUC compared to the Random Forest (0.88 vs 0.91).

Overall, our optimal classifiers, for both brain compression and edema, vastly outperformed provider documentation in sensitivity due its ability to more readily classify true brain compression and brain edema cases. We do see, however, a lower specificity in our brain compression classifiers when compared to providers (0.80 vs 0.95), but this can likely be attributed to the fact that provider documentation overwhelmingly failed to document brain compression. This lead to a high number of true negatives and low number of false positives resulting in a very high specificity. A detailed metric comparison of our classifiers' performance compared to provider documentation performance is presented In **Table 4A and 4B**.

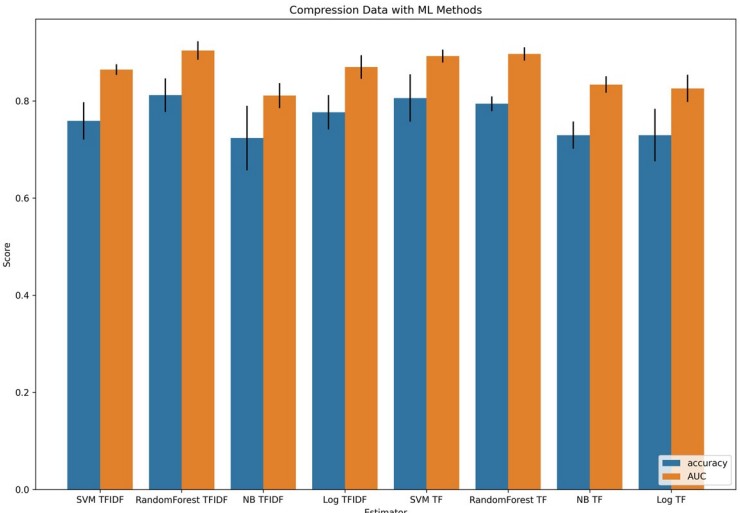
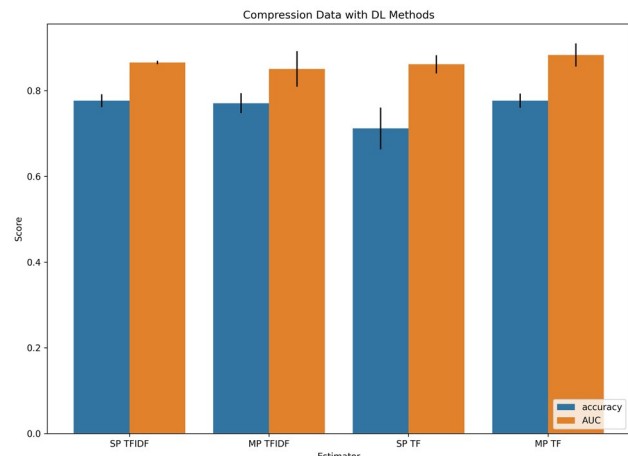

**Fig 2. Machine learning and deep learning model performance on brain compression data.** (A) Machine learning classifiers' performance methods with both frequency (TF) and term frequency-inverse document frequency (TFIDF) tokenization strategies. (B) Deep learning classifiers' performance methods with both frequency and term frequency-inverse document frequency (TFIDF) tokenization strategies. SVM = support vector machine; NB = Naïve bayes; Log = Logistic regression.

## Discussion

In contemporary American healthcare, the benefits of improved documentation are at best infrequently and indirectly apparent to those on whom the burden of documentation falls. As such, despite the longevity of the DRG-based reimbursement system, sporadic hospital- and practice-based efforts to optimize inpatient documentation abound [1, 2, 4–6, 10, 14, 21–29]. Given the relatively large financial impact of neurosurgical procedures to overall hospital finances and the significant costs of non-operative trauma care, developing simple, reproducible, and efficacious mechanisms for documentation improvement for inpatient neurosurgical practitioners is of paramount importance [2, 5, 30]. In this context, we report the successful

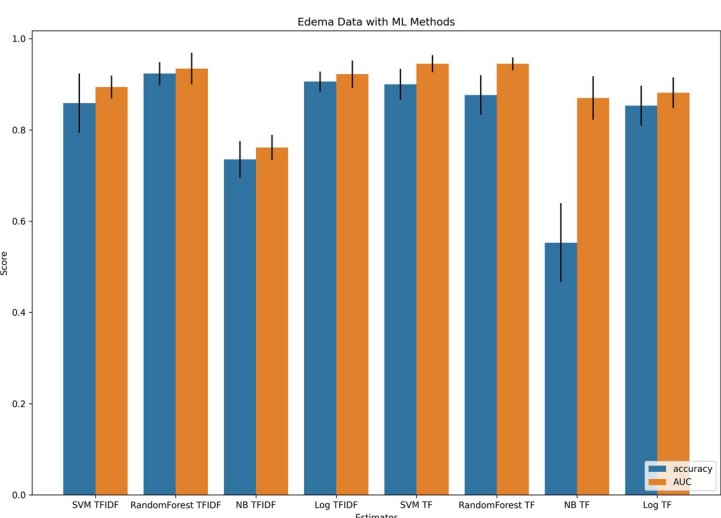
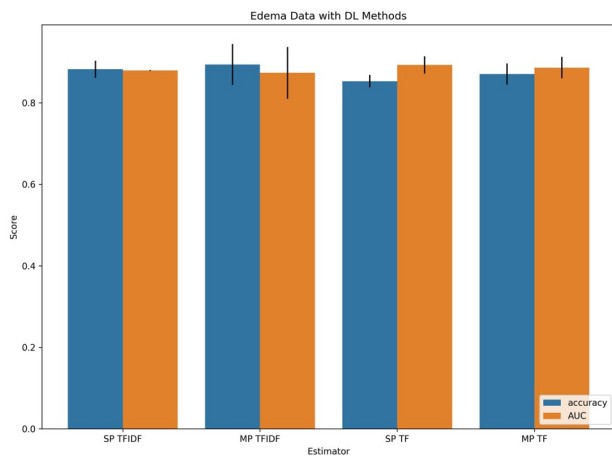

**Fig 3. Machine learning and deep learning model performance on brain edema data.** (A) Machine learning classifiers' performance methods with both frequency (TF) and term frequency-inverse document frequency (tfidf) tokenization strategies. (B) Deep learning classifiers' performance methods with both frequency and term frequency-inverse document frequency (TFIDF) tokenization strategies. SVM = support vector machine; NB = Naïve bayes; Log = Logistic regression.

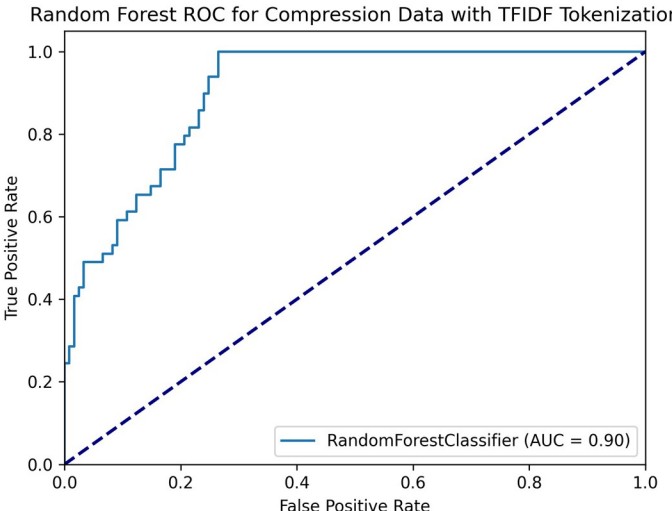

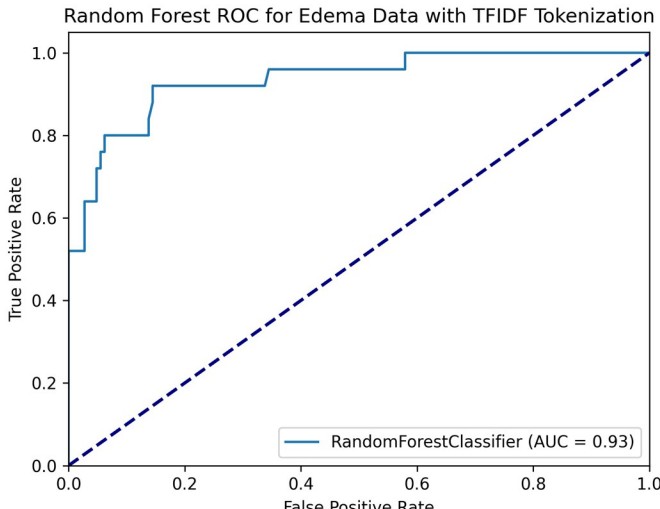

**Fig 4. Receiver operating characteristic (ROC) curves for random forest classifier with TF-IDF tokenization.** (A) Estimator trained for brain compression classification. (B) Estimator trained for brain edema classification. AUC = area under the curve.

development and validation of an NLP/ML-based algorithm for the identification of two common neurosurgical CC/MCC's from the reports of CTs or MRIs of the brain. When assessed against real-life performance of inpatient neurosurgical providers, our algorithm outperformed baseline provider documentation after the recent implementation of a documentation improvement effort. These results suggest that ML-based decision support should be considered as efficient and cost-effective components of future documentation improvement efforts and, in this specific context, could suggest diagnoses that could be documented along with diagnosis-specific treatment plans. More broadly, the implementation of an efficient, text-based algorithm could have many applications to inpatient care outside of neurosurgery alone.

Time spent documenting in EMR already consumes multiple hours in the average surgical workday [9, 31]. This documentation burden is increasingly significant for inpatient medical and surgical residents, who, along with inpatient APPs, perform the majority of consequential documentation for hospital inpatients in academic centers [7, 8, 14, 23, 31, 32]. Surgeon perception that additional documentation may not be clinically meaningful necessarily limits the implementation of documentation improvement programs, nearly all of which require investment in the form of time, personnel, or both [33]. As previously noted, many previous interventions have coupled targeted provider education sessions with ongoing chart review to provide providers with feedback or to generate further documentation queries [2–6, 14, 21, 24, 25, 27, 28]. For instance, Fox *et al* report a cost greater than $350,000 and return on investment of 220% for a program that involved personalized documentation teaching sessions and allocation of documentation specialists to round with a trauma surgery team and to review notes at a Level 1 trauma center [10]. Similarly, for a similar intervention, Spurgeon *et al* reported that nurses working 10–15 hours/week on documentation improvement were only able to review less than half of inpatient neurosurgical notes over an 8-week time period [5]. Efforts to minimize time investment required by providers to update notes underpinned the development of the documentation query, in which a provider need only respond "yes" or "no" for the presence or absence of a given diagnosis [34]; however, even with simple systems, time-consuming manual review by documentation experts is still required to generate queries. The progress note template, which standardizes common comorbidities during documentation efforts, is another low-cost documentation improvement intervention, though contemporary success of

**Table 3.** **A.** Performance Metrics for ML and DL Models on Brain Compression. **B.** Performance Metrics for ML and DL Models on Brain Edema.

| Model | Tokenization | Compression Performance | | | | | |
|---|---|---|---|---|---|---|---|
| ML | TF-IDF | Acc | AUC | Specificity | Sensitivity | PPV | NPV |
| SVM | TF-IDF | 0.76 | 0.86 | 0.97 | 0.24 | 0.75 | 0.76 |
| RF | TF-IDF | 0.81 | 0.90 | 0.88 | 0.65 | 0.68 | 0.86 |
| NB | TF-IDF | 0.72 | 0.81 | 0.92 | 0.24 | 0.55 | 0.75 |
| Log | TF-IDF | 0.78 | 0.87 | 0.81 | 0.69 | 0.60 | 0.87 |
| ML | TF | Acc | AUC | Specificity | Sensitivity | PPV | NPV |
| SVM | TF | 0.81 | 0.89 | 0.87 | 0.65 | 0.67 | 0.86 |
| RF | TF | 0.79 | 0.90 | 0.88 | 0.57 | 0.67 | 0.84 |
| NB | TF | 0.73 | 0.83 | 0.68 | 0.86 | 0.52 | 0.92 |
| Log | TF | 0.73 | 0.83 | 0.98 | 0.12 | 0.67 | 0.73 |
| DL | TF-IDF | Acc | AUC | Specificity | Sensitivity | PPV | NPV |
| SP | TF-IDF | 0.78 | 0.87 | 0.77 | 0.80 | 0.58 | 0.90 |
| MP | TF-IDF | 0.77 | 0.85 | 0.79 | 0.73 | 0.58 | 0.88 |
| DL | TF | Acc | AUC | Specificity | Sensitivity | PPV | NPV |
| SP | TF | 0.71 | 0.86 | 1.0 | 0.0 | 0.0 | 0.71 |
| MP | TF | 0.78 | 0.88 | 0.77 | 0.80 | 0.58 | 0.90 |
| Model | Tokenization | Edema Performance | | | | | |
| ML | TF-IDF | Acc | AUC | Specificity | Sensitivity | PPV | NPV |
| SVM | TF-IDF | 0.86 | 0.89 | 1.0 | 0.04 | 1.0 | 0.86 |
| RF | TF-IDF | 0.92 | 0.94 | 1.0 | 0.48 | 1.0 | 0.91 |
| NB | TF-IDF | 0.74 | 0.76 | 0.77 | 0.56 | 0.29 | 0.91 |
| Log | TF-IDF | 0.91 | 0.92 | 0.97 | 0.52 | 0.76 | 0.92 |
| ML | TF | Acc | AUC | Specificity | Sensitivity | PPV | NPV |
| SVM | TF | 0.90 | 0.95 | 1.0 | 0.32 | 1.0 | 0.90 |
| RF | TF | 0.88 | 0.94 | 1.0 | 0.16 | 1.0 | 0.87 |
| NB | TF | 0.55 | 0.87 | 0.49 | 0.92 | 0.24 | 0.97 |
| Log | TF | 0.85 | 0.88 | 1.0 | 0.0 | 0.0 | 0.85 |
| DL | TF-IDF | Acc | AUC | Specificity | Sensitivity | PPV | NPV |
| SP | TF-IDF | 0.88 | 0.88 | 0.97 | 0.40 | 0.67 | 0.90 |
| MP | TF-IDF | 0.89 | 0.87 | 0.98 | 0.40 | 0.77 | 0.90 |
| DL | TF | Acc | AUC | Specificity | Sensitivity | PPV | NPV |
| SP | TF | 0.85 | 0.89 | 1.0 | 0.0 | 0.0 | 0.85 |
| MP | TF | 0.87 | 0.89 | 0.95 | 0.40 | 0.59 | 0.90 |

SVM = support vector machine; RF = Random Forest; NB = Naïve bayes; Log = Logistic regression; Acc = Accuracy; AUC = Area Under the Curve; PPV = Positive Predictive Value; NPV = Negative Predictive Value

simple, paper-based checklists have, perplexingly, been shown to yield more thorough documentation than EMR-based approaches [29]. In clinical contexts, documentation of "brain compression" and "brain edema" can only be reliably extracted from neuroimaging and rarely convey meaningful clinical information relative to more commonly used expressions; as such, ML-based approaches to extract these diagnoses, which are both common and commonly undocumented, may yield significant benefits relative to low costs.

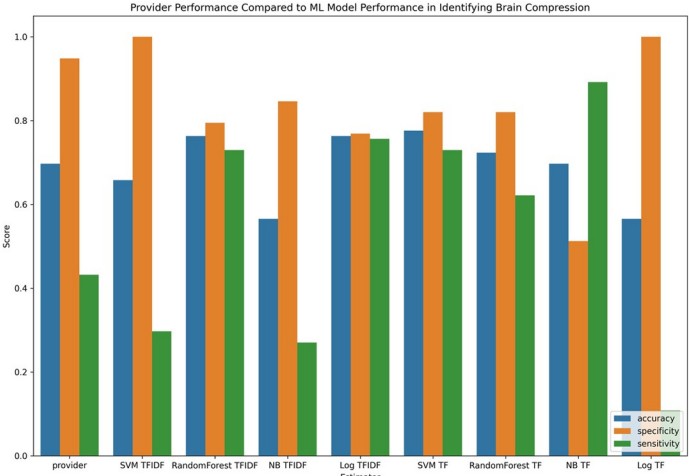
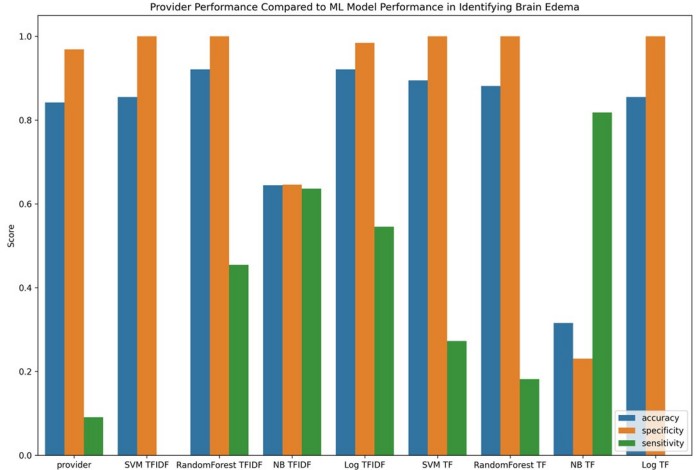

**Fig 5. Machine learning estimator and provider documentation comparison.** (A) Estimators for compression dataset. (B) Estimators for edema dataset. SVM = support vector machine; NB = Naïve bayes; Log = Logistic regression.

ML and NLP applications in neurosurgery and neuroimaging are numerous and varied [15, 35–39] and have only increased in breadth in depth since the widespread popularization of large language models (LLMs) such as ChatGPT [40–44]. A variety of methods utilizing either radiology reports, raw images, or both, have been successfully applied in a variety of clinical applications [45, 46]. Decision support for clinical documentation may offer a particularly fruitful application of these technologies, especially given that the imperative is to augment documentation at provider discretion without necessarily changing the course of patient care. Documentation efforts likely require a flexible approach for ML applications–certain diagnoses, such as "brain compression", can be exclusively learned through imaging reports. Others, such as "encephalopathy", cannot and instead would require parsing provider documentation and medication administration. Another challenge of clinical documentation efforts is that documentation requirements for various stakeholders may not necessarily overlap and, furthermore, may change over time with the release of new documentation standards. A final challenge, which will likely become more prominent with the availability of public LLMs, is the requirement to protect patient data confidentiality [47]. For instance, while an internally developed algorithm such as our own may not jeopardize patient health information (PHI) as both the training and implementation of the model are local; however, use of public LLMs may easily risk transmission of PHI to servers of an external organization. Future applications of these technologies will need to be aware of these particular risks. Nevertheless, the opportunities for NLP are significant and likely extend beyond comorbidity documentation to clinical decision support, safety oversight, telehealth, clinical encounter documentation, and informed patient consent, among many others.

While this project does demonstrate the feasibility of NLP-based decision support for clinical neurosurgical documentation, it does have notable limitations. Our optimal random forest classifiers demonstrated relatively low sensitivity (0.65 and 0.48, for compression and edema, respectively) relative to their high specificity (0.88 and 1.0, for compression and edema respectively). However, for a clinical decision support system, a high specificity in the context of a lower sensitivity is preferred to low specificity and high sensitivity, as an optimal clinical decision support system should generate few false positives and a high number of true positives. Our lower sensitivity numbers are likely attributable to the use of a more naïve NLP approach by only looking at the presence of individual word tokens rather than processing and

**Table 4.** **A.** Estimator and Provider Performance Comparison for Brain Compression. **B.** Estimator and Provider Performance Comparison for Brain Compression.

| Model | Tokenization | Compression Performance | | | | | |
|---|---|---|---|---|---|---|---|
| Provider | NA | Acc | AUC | Specificity | Sensitivity | PPV | NPV |
| Provider | NA | 0.70 | NA | 0.95 | 0.43 | 0.89 | 0.64 |
| ML | TF-IDF | Acc | AUC | Specificity | Sensitivity | PPV | NPV |
| SVM | TF-IDF | 0.66 | 0.88 | 1.00 | 0.30 | 1.00 | 0.60 |
| RF | TF-IDF | 0.76 | 0.89 | 0.80 | 0.73 | 0.77 | 0.76 |
| NB | TF-IDF | 0.57 | 0.74 | 0.85 | 0.27 | 0.63 | 0.55 |
| Log | TF-IDF | 0.76 | 0.87 | 0.77 | 0.76 | 0.76 | 0.77 |
| ML | TF | Acc | AUC | Specificity | Sensitivity | PPV | NPV |
| SVM | TF | 0.78 | 0.88 | 0.82 | 0.73 | 0.79 | 0.76 |
| RF | TF | 0.72 | 0.89 | 0.82 | 0.62 | 0.77 | 0.70 |
| NB | TF | 0.70 | 0.74 | 0.51 | 0.89 | 0.63 | 0.83 |
| Log | TF | 0.57 | 0.79 | 1.00 | 0.11 | 1.0 | 0.54 |
| DL | TF-IDF | Acc | AUC | Specificity | Sensitivity | PPV | NPV |
| SP | TF-IDF | 0.74 | 0.82 | 0.67 | 0.81 | 0.70 | 0.79 |
| MP | TF-IDF | 0.75 | 0.83 | 0.72 | 0.78 | 0.73 | 0.78 |
| DL | TF | Acc | AUC | Specificity | Sensitivity | PPV | NPV |
| SP | TF | 0.51 | 0.82 | 1.00 | 0.00 | 0.00 | 0.51 |
| MP | TF | 0.74 | 0.84 | 0.67 | 0.81 | 0.70 | 0.79 |
| Model | Tokenization | Edema Performance | | | | | |
| Provider | NA | Acc | AUC | Specificity | Sensitivity | PPV | NPV |
| Provider | NA | 0.84 | NA | 0.97 | 0.09 | 0.33 | 0.86 |
| ML | TF-IDF | Acc | AUC | Specificity | Sensitivity | PPV | NPV |
| SVM | TF-IDF | 0.86 | 0.87 | 1.00 | 0.00 | 0.00 | 0.86 |
| RF | TF-IDF | 0.92 | 0.91 | 1.00 | 0.45 | 1.00 | 0.92 |
| NB | TF-IDF | 0.64 | 0.68 | 0.65 | 0.64 | 0.23 | 0.91 |
| Log | TF-IDF | 0.92 | 0.88 | 0.98 | 0.54 | 0.86 | 0.93 |
| ML | TF | Acc | AUC | Specificity | Sensitivity | PPV | NPV |
| SVM | TF | 0.89 | 0.88 | 1.00 | 0.27 | 1.00 | 0.89 |
| RF | TF | 0.88 | 0.91 | 1.00 | 0.18 | 1.00 | 0.88 |
| NB | TF | 0.32 | 0.70 | 0.23 | 0.82 | 0.15 | 0.88 |
| Log | TF | 0.86 | 0.71 | 1.00 | 0.00 | 0.00 | 0.86 |
| DL | TF-IDF | Acc | AUC | Specificity | Sensitivity | PPV | NPV |
| SP | TF-IDF | 0.88 | 0.84 | 0.95 | 0.45 | 0.63 | 0.91 |
| MP | TF-IDF | 0.89 | 0.83 | 0.97 | 0.45 | 0.71 | 0.91 |
| DL | TF | Acc | AUC | Specificity | Sensitivity | PPV | NPV |
| SP | TF | 0.86 | 0.76 | 1.00 | 0.00 | 0.00 | 0.86 |
| MP | TF | 0.86 | 0.81 | 0.94 | 0.36 | 0.50 | 0.90 |

SVM = support vector machine; RF = Random Forest; NB = Naïve bayes; Log = Logistic regression; Acc = Accuracy; AUC = Area Under the Curve; PPV = Positive Predictive Value; NPV = Negative Predictive Value

interpreting word tokens in context of the report as a whole. Future studies should focus on increasing sensitivity, which would likely occur with a larger dataset and the use of more intricate NLP infrastructures such as recursive neural networks or transformers that can more

readily contextualize blocks of text as a whole. From the perspective of data collection, the development of this particular NLP model was based on radiology reports generated within a single health care system; as such, its applicability to different reporting systems or in other languages may be limited. Furthermore, the reports were reviewed by a single author. As previously noted, this effort evaluated the diagnosis of two particular comorbidities that could be readily ascertained from neuroimaging. Finally, tactful EMR implementation will be necessary to present results of this algorithm to clinicians in a way that encourages responses and meaningfully improves clinical documentation.

## Conclusions

An NLP-based ML algorithm can reliably detect 2 major comorbidities for neurosurgical patients from radiology reports. Algorithm performance exceeds real-life documentation performance.

## Supporting information

**S1 File.**
(ZIP)

**S1 Dataset.**
(ZIP)

## Author Contributions

**Conceptualization:** Rahul A. Sastry, Aayush Setty, David D. Liu, Bryan Zheng, Rohaid Ali, Robert J. Weil, G. Dean Roye, Curtis E. Doberstein, Adetokunbo A. Oyelese, Tianyi Niu, Ziya L. Gokaslan, Albert E. Telfeian.

**Data curation:** Rahul A. Sastry.

**Formal analysis:** Rahul A. Sastry, Aayush Setty.

**Investigation:** Rahul A. Sastry.

**Methodology:** Rahul A. Sastry, David D. Liu, Bryan Zheng, Robert J. Weil.

**Project administration:** Rahul A. Sastry, David D. Liu, Bryan Zheng.

**Resources:** Rahul A. Sastry, David D. Liu.

**Software:** Rahul A. Sastry, Aayush Setty, Bryan Zheng.

**Supervision:** Rahul A. Sastry, David D. Liu, Bryan Zheng, Robert J. Weil.

**Validation:** Rahul A. Sastry, Aayush Setty, David D. Liu.

**Visualization:** Aayush Setty.

**Writing – original draft:** Rahul A. Sastry, Aayush Setty, David D. Liu, Bryan Zheng, Rohaid Ali, Robert J. Weil.

**Writing – review & editing:** Rahul A. Sastry, Aayush Setty, David D. Liu, Bryan Zheng, Rohaid Ali, Robert J. Weil, G. Dean Roye, Curtis E. Doberstein, Adetokunbo A. Oyelese, Tianyi Niu, Ziya L. Gokaslan, Albert E. Telfeian.

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
