## [Decision Letter · Decision Letter 0]

19 Oct 2022

PONE-D-22-22485Natural Language Processing Augments Comorbidity Documentation in Neurosurgical Inpatient AdmissionsPLOS ONE

Dear Dr. Sastry,

Thank you for submitting your manuscript to PLOS ONE. After careful consideration, we feel that it has merit but does not fully meet PLOS ONE’s publication criteria as it currently stands. Therefore, we invite you to submit a revised version of the manuscript that addresses the points raised during the review process.

We look forward to receiving your revised manuscript.

Kind regards,

Vijayalakshmi Kakulapati, Ph.D

Academic Editor

PLOS ONE

Journal Requirements:

2. Please ensure you state in the Methods section of your manuscript text all relevant information regarding participant consent. This is in addition to the information you have provided in the Ethics Statement section of the online submission form.

Additionally, please note that PLOS ONE has specific guidelines on code sharing for submissions in which author-generated code underpins the findings in the manuscript. In these cases, all author-generated code must be made available without restrictions upon publication of the work. Please review our guidelines at https://journals.plos.org/plosone/s/materials-and-software-sharing#loc-sharing-code and ensure that your code is shared in a way that follows best practice and facilitates reproducibility and reuse.

Reviewers' comments:

Reviewer's Responses to Questions

**Comments to the Author**

1. Is the manuscript technically sound, and do the data support the conclusions?

Reviewer #1: Partly

Reviewer #2: Yes

2. Has the statistical analysis been performed appropriately and rigorously? 

Reviewer #1: Yes

Reviewer #2: Yes

3. Have the authors made all data underlying the findings in their manuscript fully available?

Reviewer #1: Yes

Reviewer #2: No

4. Is the manuscript presented in an intelligible fashion and written in standard English?

Reviewer #1: Yes

Reviewer #2: Yes

5. Review Comments to the Author

Reviewer #1: The authors present a valuable solution to document augmentation through use of ML algorithms, by inferring two comorbidities from the text accompanying patient brain scan (MRI, and CT). While I value and command the authors initiative, time, and effort in conducting this research, I do hope that including a note on the following considerations will improve the quality of the work.

Main considerations about the current manuscript are as follows:

(1) The data need to be described in more detail. The authors have trained ML models for two binary classification problems, effectively rendering four different classes (non-compression vs compression) (non-edema vs edema). However the membership size in each of these classes is not clear. The authors mention that they have collected data both from neurosurgery department as well as from emergency to include both positive and negative controls, however they do not specify how many positive/negative examples were present in the ML datasets. The difference in these numbers, referred to as “class imbalance” can negatively affect the performance of the model and can cause a one-sided optimization, by improving specificity only, disregarding sensitivity (or vice versa). In the presence of class imbalance, there are also methods to deal with this problem that the authors could utilize. However, since these class balances are not specified in the current manuscript it is not possible to come to a conclusion.

(2) Also related to (1), the metric that the ML model is optimized for can be of significance. While it is not clearly specified in the document, I am inferring that the optimization metric (loss function) used for the ML training has been the accuracy value. In certain scenarios, especially in the presence of class imbalance, optimizing for higher accuracy might not be the best choice. Regardless of whether there was imbalance or not, choice of loss function is of importance and has to be specified for reproducible result.

(3) Again not unrelated to (1), it appears that the training and test sets have been collected using a different protocol than the validation set. Specifically, the validation set is only based on neurosurgery patients while the training and test sets include patients from the emergency department as well. The grid search performed to select the best parameters is generally performed on the validation set which should have the same distribution as the training and test sets. Is there evidence that the distribution is not different among validation and train/test sets.

(4) The authors have described their method for tokenization, however some statistics or a few examples could be helpful to both, better understand the problem and have a better picture of the solution. In particular, is it possible that some of the texts accompanying the MRI/CT images do already mention brain edema/compression. If so, what is the percentage of these images, and what is the benefit of using an ML models in such texts? Wouldn’t limiting the training examples to ones without such clear indication result in better performing algorithms?

And a final minor point:

(5) “After” (I assume) is missing in “during and after admission” on line 84.

Reviewer #2: Framing of the abstract: NLP cannot be applied to images. I would reframe it to state that “948 CT or MRI scans’ radiological reports were examined” or something of the sort (instead of "we applied NLP to images", which isn't possible). It’s important to be clear if you’re talking about the reports or images because in other instances where RAS reviews the images, it’s unclear if he’s looking at reports or if he’s looking at the images themselves.

Data-preprocessing: Few small points of clarification. 1) How did you split word tokens. Based on white-space?

Data Availability: It’s understandable that there are restrictions placed on data access. However, I think it might be useful for those looking to recreate the approaches on their own data if the code was made available. It would also help clear up any confusion about pre-processing.

Results: Reporting accuracy is not particularly meaningful if I don’t know the prevalence. If brain compression only occurs in 10% of patients, and I predict “no brain compression” for all patients, my accuracy is 90% but my predictions are useless. It’s fine to use accuracy as the single measure reported in the text, but in the figure/appendix you should be reporting a wider array of measures. At minimum: precision/ppv, recall/sensitivity, npv, specificity.

I’m not sure why this is the case, (could be on my side), but the images are very poor in quality. It’s very hard to make out what’s written.

Reporting and comparing the AUC for the provider thresholds is likely meaningless/incorrect. The AUC procedure involves testing the performance of a classifier at different thresholds. However, the idea of thresholds doesn’t apply to human documentation? Again, the comments about reporting more/different metrics from above applies here.

I do see some numbers sensitivity/specific in the discussion, but this should be upfront in the results section not in the discussion. It would also be interesting to explore why the sensitivity is so low and discuss the sensitivity of the humans vs machine.

6. PLOS authors have the option to publish the peer review history of their article (what does this mean?). If published, this will include your full peer review and any attached files.

Reviewer #1: No

Reviewer #2: No

---

## [Author Response · Author response to Decision Letter 0]

3 Dec 2022

Rahul Sastry, MD

Department of Neurosurgery

Brown University 

593 Eddy St, APC-6

Providence, RI 02903

rahul.sastry@gmail.com

November 28, 2022

Dear Dr. Kakulapati,

Thank you for providing a thoughtful review of our manuscript. We have attempted to clarify all of the points raised by the reviewers as follows: 

(1) The data need to be described in more detail. The authors have trained ML models for two binary classification problems, effectively rendering four different classes (non-compression vs compression) (non-edema vs edema). However the membership size in each of these classes is not clear. The authors mention that they have collected data both from neurosurgery department as well as from emergency to include both positive and negative controls, however they do not specify how many positive/negative examples were present in the ML datasets. The difference in these numbers, referred to as “class imbalance” can negatively affect the performance of the model and can cause a one-sided optimization, by improving specificity only, disregarding sensitivity (or vice versa). In the presence of class imbalance, there are also methods to deal with this problem that the authors could utilize. However, since these class balances are not specified in the current manuscript it is not possible to come to a conclusion.

(2) Also related to (1), the metric that the ML model is optimized for can be of significance. While it is not clearly specified in the document, I am inferring that the optimization metric (loss function) used for the ML training has been the accuracy value. In certain scenarios, especially in the presence of class imbalance, optimizing for higher accuracy might not be the best choice. Regardless of whether there was imbalance or not, choice of loss function is of importance and has to be specified for reproducible result.

We made our methods section clearer on the structure of our datasets and how they were used. One area of confusion we may have introduced is the nature of our “validation set”. This validation set (now renamed to provider comparison dataset) is a subset of our test set that has data on provider documentation. We used this subset of the test set to compare the performance of our model to providers at our affiliated institution. There was no hyperparameter tuning done with the use of the set but rather the tuning was done within the training set itself with the use of 5-fold cross validation. We also added a note about how word tokens were generated – we split on white space and subsequently removed punctuation and stop words.

To your concerns on data description and class imbalance, we have included an additional table describing the class proportions in both the brain compression and brain edema training and testing datasets. To make sure that our classifiers were not biased by the slightly imbalanced nature of our datasets, we conducted hyperparameter tuning with respect to ROC AUC to prevent bias from accuracy optimization. The metric/loss function that the classifiers used to optimize their own internal parameters varied from estimator to estimator as some have their own special loss functions whereas some used cross entropy loss. Brief descriptions of these loss functions have been added to the methods section of the manuscript.

(3) Again not unrelated to (1), it appears that the training and test sets have been collected using a different protocol than the validation set. Specifically, the validation set is only based on neurosurgery patients while the training and test sets include patients from the emergency department as well. The grid search performed to select the best parameters is generally performed on the validation set which should have the same distribution as the training and test sets. Is there evidence that the distribution is not different among validation and train/test sets.

(4) The authors have described their method for tokenization, however some statistics or a few examples could be helpful to both, better understand the problem and have a better picture of the solution. In particular, is it possible that some of the texts accompanying the MRI/CT images do already mention brain edema/compression. If so, what is the percentage of these images, and what is the benefit of using an ML models in such texts? Wouldn’t limiting the training examples to ones without such clear indication result in better performing algorithms?

Reviewer #1 made a very interesting point about the prevalence of the word tokens “compression” and “edema” within the radiology reports that may render the use of ML and DL classifiers moot. To this point, we conducted a sweep through the data and found that, firstly, the presence of the tokens “compression” and “edema” were relatively low within the corpus (1.7% and 15.5%) and, secondly, the presence of these tokens did not necessarily correlate deterministically with a positive label (27% and 54% of reports with compression and edema tokens were positive for compression and edema respectively). Therefore, we believe it is valuable to keep these records in the dataset as the presence of these tokens in conjunction with other specific tokens may be a valuable relationship for the classifiers to learn. In addition, we added a figure showing how frequency tokenization and TF-IDF tokenization vectorizes corpuses in a small example to help readers understand the merit of each approach.

(5) “After” (I assume) is missing in “during and after admission” on line 84.

This error has been corrected. 

Framing of the abstract: NLP cannot be applied to images. I would reframe it to state that “948 CT or MRI scans’ radiological reports were examined” or something of the sort (instead of "we applied NLP to images", which isn't possible). It’s important to be clear if you’re talking about the reports or images because in other instances where RAS reviews the images, it’s unclear if he’s looking at reports or if he’s looking at the images themselves.

Few small points of clarification. 1) How did you split word tokens. Based on white-space?

As noted above, tokens were split on white space. 

It’s understandable that there are restrictions placed on data access. However, I think it might be useful for those looking to recreate the approaches on their own data if the code was made available. It would also help clear up any confusion about pre-processing.

We also have made anonymized versions of our datasets available as well as our code scripts.

Reporting accuracy is not particularly meaningful if I don’t know the prevalence. If brain compression only occurs in 10% of patients, and I predict “no brain compression” for all patients, my accuracy is 90% but my predictions are useless. It’s fine to use accuracy as the single measure reported in the text, but in the figure/appendix you should be reporting a wider array of measures. At minimum: precision/ppv, recall/sensitivity, npv, specificity.

We also expounded on our reported metrics in the results section where we included new tables showing accuracy, AUC, specificity, sensitivity, PPV, and NPV for every classifier as well as the providers (excluding AUC). Provider AUC was also removed due to it likely being an incorrect metric in context of provider performance and instead replaced its analysis with specificity. 

I’m not sure why this is the case, (could be on my side), but the images are very poor in quality. It’s very hard to make out what’s written.

The images were generated again and converted to the appropriate format through a different process to ensure clarity. We also have made anonymized versions of our datasets available as well as our code scripts.

Reporting and comparing the AUC for the provider thresholds is likely meaningless/incorrect. The AUC procedure involves testing the performance of a classifier at different thresholds. However, the idea of thresholds doesn’t apply to human documentation? Again, the comments about reporting more/different metrics from above applies here.

I do see some numbers sensitivity/specific in the discussion, but this should be upfront in the results section not in the discussion. It would also be interesting to explore why the sensitivity is so low and discuss the sensitivity of the humans vs machine.

These concerns have been addressed, as above.

Thank you again for your time reviewing our manuscript, these edits have strengthened our manuscript and we hope that we have addressed your comments adequately. We would be happy to provide additional clarification or revisions as needed.

Sincerely,

Aayush Setty

Rahul Sastry, MD

---

## [Decision Letter · Decision Letter 1]

12 Dec 2022

PONE-D-22-22485R1Natural Language Processing Augments Comorbidity Documentation in Neurosurgical Inpatient AdmissionsPLOS ONE

Dear Dr. Sastry,

Thank you for submitting your manuscript to PLOS ONE. After careful consideration, we feel that it has merit but does not fully meet PLOS ONE’s publication criteria as it currently stands. Therefore, we invite you to submit a revised version of the manuscript that addresses the points raised during the review process.

We look forward to receiving your revised manuscript.

Kind regards,

Vijayalakshmi Kakulapati, Ph.D

Academic Editor

PLOS ONE

Journal Requirements:

Reviewers' comments:

Reviewer #2: The paper’s presentation is much improved. I have only a few comments:

> TF-IDF and frequency should be TF-IDF and TF. TF stands for term frequency which is just vanilla bag of words.

> Line 142 is incomplete.

> It is not clear how the predictions for the provider sample are arrived at. Are the classification of “Provider Model” the terms used in the discharge notes? That is, the NLP is applied to the radiology report and you check the agreement with the discharge notes. If so, I think this can be stated much more clearly than it is currently done.

> “All 1,055 patient images were reviewed by a single author (RAS)”. It’s unclear why this was done. You state “to assess for the presence or absence of BC/BE”. If that is the case, are these your gold labels? Or are the gold labels extracted from the reports? If these are your gold labels then it’s poor form to have only a single person judge the entire set of images because agreement between radiologists can sometimes be quite low and this would provide meaningful context to model comparisons. Not that you should change this, you only need to highlight this as a limitation. But more importantly, it should be much more clear.

> In fact, I recommend splitting the materials section into subsections: > Gold Labels, > Machine Prediction, > Human prediction, > Preprocessing, etc. This would make it clearer when you’ve moved from one topic to another.

> Images are blurry, though it’s not clear if that’s because of the Authors or because of the submission system.

---

## [Author Response · Author response to Decision Letter 1]

30 Jan 2023

Rahul Sastry, MD

Department of Neurosurgery

Brown University 

593 Eddy St, APC-6

Providence, RI 02903

rahul.sastry@gmail.com

January 3, 2023

Dear Dr. Kakulapati,

Thank you for providing a thoughtful review of our manuscript. We have attempted to clarify all of the points raised by the reviewers as follows: 

> TF-IDF and frequency should be TF-IDF and TF. TF stands for term frequency which is just vanilla bag of words.

These abbreviations have been updated in the manuscript and in the figures

> Line 142 is incomplete.

This line has been corrected.

> It is not clear how the predictions for the provider sample are arrived at. Are the classification of “Provider Model” the terms used in the discharge notes? That is, the NLP is applied to the radiology report and you check the agreement with the discharge notes. If so, I think this can be stated much more clearly than it is currently done.

The gold standards for diagnoses were arrived at via author review of relevant imaging. The actual provider performance was assessed by successful inclusion or exclusion of terms in the discharge summary. The sentence has been updated:

Records for patients in the provider comparison cohort, the temporal range of which was chosen to reflect documentation practices after successful implementation of a provider-education intervention in late 2020, were also manually queried for discharge summary documentation of “brain compression” and “brain edema”; as such, for patients in this cohort, presence or absence of either term in the discharge summary were used to assess the performance of real-world provider documentation against the gold standard of author review.

> “All 1,055 patient images were reviewed by a single author (RAS)”. It’s unclear why this was done. You state “to assess for the presence or absence of BC/BE”. If that is the case, are these your gold labels? Or are the gold labels extracted from the reports? If these are your gold labels then it’s poor form to have only a single person judge the entire set of images because agreement between radiologists can sometimes be quite low and this would provide meaningful context to model comparisons. Not that you should change this, you only need to highlight this as a limitation. But more importantly, it should be much more clear.

> In fact, I recommend splitting the materials section into subsections: > Gold Labels, > Machine Prediction, > Human prediction, > Preprocessing, etc. This would make it clearer when you’ve moved from one topic to another.

The images were indeed reviewed by a single author. We have acknowledged this limitation in the discussion. As suggested, additional sub-labels have been added to the methods section. 

> Images are blurry, though it’s not clear if that’s because of the Authors or because of the submission system.

Figures were verified with PACE prior to submission.

For the editors’ reference, the previously submitted responses to reviewers are included below:

(1) The data need to be described in more detail. The authors have trained ML models for two binary classification problems, effectively rendering four different classes (non-compression vs compression) (non-edema vs edema). However the membership size in each of these classes is not clear. The authors mention that they have collected data both from neurosurgery department as well as from emergency to include both positive and negative controls, however they do not specify how many positive/negative examples were present in the ML datasets. The difference in these numbers, referred to as “class imbalance” can negatively affect the performance of the model and can cause a one-sided optimization, by improving specificity only, disregarding sensitivity (or vice versa). In the presence of class imbalance, there are also methods to deal with this problem that the authors could utilize. However, since these class balances are not specified in the current manuscript it is not possible to come to a conclusion.

(2) Also related to (1), the metric that the ML model is optimized for can be of significance. While it is not clearly specified in the document, I am inferring that the optimization metric (loss function) used for the ML training has been the accuracy value. In certain scenarios, especially in the presence of class imbalance, optimizing for higher accuracy might not be the best choice. Regardless of whether there was imbalance or not, choice of loss function is of importance and has to be specified for reproducible result.

We made our methods section clearer on the structure of our datasets and how they were used. One area of confusion we may have introduced is the nature of our “validation set”. This validation set (now renamed to provider comparison dataset) is a subset of our test set that has data on provider documentation. We used this subset of the test set to compare the performance of our model to providers at our affiliated institution. There was no hyperparameter tuning done with the use of the set but rather the tuning was done within the training set itself with the use of 5-fold cross validation. We also added a note about how word tokens were generated – we split on white space and subsequently removed punctuation and stop words.

To your concerns on data description and class imbalance, we have included an additional table describing the class proportions in both the brain compression and brain edema training and testing datasets. To make sure that our classifiers were not biased by the slightly imbalanced nature of our datasets, we conducted hyperparameter tuning with respect to ROC AUC to prevent bias from accuracy optimization. The metric/loss function that the classifiers used to optimize their own internal parameters varied from estimator to estimator as some have their own special loss functions whereas some used cross entropy loss. Brief descriptions of these loss functions have been added to the methods section of the manuscript.

(3) Again not unrelated to (1), it appears that the training and test sets have been collected using a different protocol than the validation set. Specifically, the validation set is only based on neurosurgery patients while the training and test sets include patients from the emergency department as well. The grid search performed to select the best parameters is generally performed on the validation set which should have the same distribution as the training and test sets. Is there evidence that the distribution is not different among validation and train/test sets.

(4) The authors have described their method for tokenization, however some statistics or a few examples could be helpful to both, better understand the problem and have a better picture of the solution. In particular, is it possible that some of the texts accompanying the MRI/CT images do already mention brain edema/compression. If so, what is the percentage of these images, and what is the benefit of using an ML models in such texts? Wouldn’t limiting the training examples to ones without such clear indication result in better performing algorithms?

Reviewer #1 made a very interesting point about the prevalence of the word tokens “compression” and “edema” within the radiology reports that may render the use of ML and DL classifiers moot. To this point, we conducted a sweep through the data and found that, firstly, the presence of the tokens “compression” and “edema” were relatively low within the corpus (1.7% and 15.5%) and, secondly, the presence of these tokens did not necessarily correlate deterministically with a positive label (27% and 54% of reports with compression and edema tokens were positive for compression and edema respectively). Therefore, we believe it is valuable to keep these records in the dataset as the presence of these tokens in conjunction with other specific tokens may be a valuable relationship for the classifiers to learn. In addition, we added a figure showing how frequency tokenization and TF-IDF tokenization vectorizes corpuses in a small example to help readers understand the merit of each approach.

(5) “After” (I assume) is missing in “during and after admission” on line 84.

This error has been corrected. 

Framing of the abstract: NLP cannot be applied to images. I would reframe it to state that “948 CT or MRI scans’ radiological reports were examined” or something of the sort (instead of "we applied NLP to images", which isn't possible). It’s important to be clear if you’re talking about the reports or images because in other instances where RAS reviews the images, it’s unclear if he’s looking at reports or if he’s looking at the images themselves.

Few small points of clarification. 1) How did you split word tokens. Based on white-space?

As noted above, tokens were split on white space. 

It’s understandable that there are restrictions placed on data access. However, I think it might be useful for those looking to recreate the approaches on their own data if the code was made available. It would also help clear up any confusion about pre-processing.

We also have made anonymized versions of our datasets available as well as our code scripts.

Reporting accuracy is not particularly meaningful if I don’t know the prevalence. If brain compression only occurs in 10% of patients, and I predict “no brain compression” for all patients, my accuracy is 90% but my predictions are useless. It’s fine to use accuracy as the single measure reported in the text, but in the figure/appendix you should be reporting a wider array of measures. At minimum: precision/ppv, recall/sensitivity, npv, specificity.

We also expounded on our reported metrics in the results section where we included new tables showing accuracy, AUC, specificity, sensitivity, PPV, and NPV for every classifier as well as the providers (excluding AUC). Provider AUC was also removed due to it likely being an incorrect metric in context of provider performance and instead replaced its analysis with specificity. 

I’m not sure why this is the case, (could be on my side), but the images are very poor in quality. It’s very hard to make out what’s written.

The images were generated again and converted to the appropriate format through a different process to ensure clarity. We also have made anonymized versions of our datasets available as well as our code scripts.

Reporting and comparing the AUC for the provider thresholds is likely meaningless/incorrect. The AUC procedure involves testing the performance of a classifier at different thresholds. However, the idea of thresholds doesn’t apply to human documentation? Again, the comments about reporting more/different metrics from above applies here.

I do see some numbers sensitivity/specific in the discussion, but this should be upfront in the results section not in the discussion. It would also be interesting to explore why the sensitivity is so low and discuss the sensitivity of the humans vs machine.

These concerns have been addressed, as above.

Thank you again for your time reviewing our manuscript, these edits have strengthened our manuscript and we hope that we have addressed your comments adequately. We would be happy to provide additional clarification or revisions as needed.

Sincerely,

Aayush Setty

Rahul Sastry, MD

---

## [Decision Letter · Decision Letter 2]

4 Oct 2023

PONE-D-22-22485R2Natural Language Processing Augments Comorbidity Documentation in Neurosurgical Inpatient AdmissionsPLOS ONE

Dear Dr. Sastry,

Thank you for submitting your manuscript to PLOS ONE. After careful consideration, we feel that it has merit but does not fully meet PLOS ONE’s publication criteria as it currently stands. Therefore, we invite you to submit a revised version of the manuscript that addresses the points raised during the review process.

We look forward to receiving your revised manuscript.

Kind regards,

Vijayalakshmi Kakulapati, Ph.D

Academic Editor

PLOS ONE 

Reviewers' comments:

Reviewer's Responses to Questions

**Comments to the Author**

1. If the authors have adequately addressed your comments raised in a previous round of review and you feel that this manuscript is now acceptable for publication, you may indicate that here to bypass the “Comments to the Author” section, enter your conflict of interest statement in the “Confidential to Editor” section, and submit your "Accept" recommendation.

Reviewer #2: All comments have been addressed

Reviewer #3: All comments have been addressed

2. Is the manuscript technically sound, and do the data support the conclusions?

Reviewer #2: Yes

Reviewer #3: Yes

3. Has the statistical analysis been performed appropriately and rigorously? 

Reviewer #2: Yes

Reviewer #3: Yes

4. Have the authors made all data underlying the findings in their manuscript fully available?

Reviewer #2: No

Reviewer #3: Yes

5. Is the manuscript presented in an intelligible fashion and written in standard English?

Reviewer #2: Yes

Reviewer #3: Yes

6. Review Comments to the Author

Reviewer #2: (No Response)

Reviewer #3: Clarity and Presentation:

The paper is well-structured and clearly presents the integration of natural language processing (NLP) for augmenting comorbidity documentation in neurosurgical inpatient admissions. The methodology and results are articulated effectively.

Originality and Contribution:

The paper addresses a significant issue in the healthcare domain, where comorbidity documentation is essential for patient care and treatment planning. The utilization of NLP to enhance this process is an innovative and valuable contribution.

Technical Soundness:

The technical aspects of implementing NLP for comorbidity documentation should be elaborated upon further. Explain the NLP techniques, tools, and algorithms used in detail. Provide insights into the data preprocessing, feature extraction, and model training procedures to ensure reproducibility.

Experimental Validation:

The paper mentions improved comorbidity documentation, but it would be beneficial to include quantitative results and metrics, such as precision, recall, and F1-score, to quantify the effectiveness of the NLP approach. A comprehensive evaluation would strengthen the paper's claims.

Data Privacy and Ethical Considerations:

Discuss the ethical considerations and data privacy safeguards implemented when handling patient data in a healthcare context. Ensure that all patient information is anonymized and de-identified to protect privacy.

Future Directions:

Conclude the paper with a discussion of potential future directions and applications of NLP in healthcare documentation beyond comorbidities. How can this technology be further extended to benefit patient care and streamline medical records?

References:

Ensure that all references are up-to-date and accurately cited. Consider including recent works related to NLP applications in healthcare for a comprehensive literature review.

7. PLOS authors have the option to publish the peer review history of their article (what does this mean?). If published, this will include your full peer review and any attached files.

Reviewer #2: No

Reviewer #3: No

---

## [Author Response · Author response to Decision Letter 2]

24 Mar 2024

Rahul Sastry, MD

Department of Neurosurgery

Brown University 

593 Eddy St, APC-6

Providence, RI 02903

rahul.sastry@gmail.com

March 24, 2024

Dear Dr. Kakulapati,

Thank you for providing a thoughtful review of our manuscript. We have attempted to clarify all of the points raised by the reviewers as follows: 

1. If the authors have adequately addressed your comments raised in a previous round of review and you feel that this manuscript is now acceptable for publication, you may indicate that here to bypass the “Comments to the Author” section, enter your conflict of interest statement in the “Confidential to Editor” section, and submit your "Accept" recommendation.

Reviewer #2: All comments have been addressed

Reviewer #3: All comments have been addressed

2. Is the manuscript technically sound, and do the data support the conclusions?

Reviewer #2: Yes

Reviewer #3: Yes

3. Has the statistical analysis been performed appropriately and rigorously? 

Reviewer #2: Yes

Reviewer #3: Yes

4. Have the authors made all data underlying the findings in their manuscript fully available?

Reviewer #2: No

Reviewer #3: Yes

5. Is the manuscript presented in an intelligible fashion and written in standard English?

Reviewer #2: Yes

Reviewer #3: Yes

6. Review Comments to the Author

Reviewer #2: (No Response)

Reviewer #3: Clarity and Presentation:

The paper is well-structured and clearly presents the integration of natural language processing (NLP) for augmenting comorbidity documentation in neurosurgical inpatient admissions. The methodology and results are articulated effectively.

Originality and Contribution:

The paper addresses a significant issue in the healthcare domain, where comorbidity documentation is essential for patient care and treatment planning. The utilization of NLP to enhance this process is an innovative and valuable contribution.

Technical Soundness:

The technical aspects of implementing NLP for comorbidity documentation should be elaborated upon further. Explain the NLP techniques, tools, and algorithms used in detail. Provide insights into the data preprocessing, feature extraction, and model training procedures to ensure reproducibility.

The methods section (page 4-8) has been greatly expanded to provide additional detail. 

Experimental Validation:

The paper mentions improved comorbidity documentation, but it would be beneficial to include quantitative results and metrics, such as precision, recall, and F1-score, to quantify the effectiveness of the NLP approach. A comprehensive evaluation would strengthen the paper's claims.

Performance metrics have also been greatly expanded as suggested. This information is found in the results section. 

Data Privacy and Ethical Considerations:

Discuss the ethical considerations and data privacy safeguards implemented when handling patient data in a healthcare context. Ensure that all patient information is anonymized and de-identified to protect privacy.

Further discussion of this point has been added to the discussion (page 17) 

Future Directions:

Conclude the paper with a discussion of potential future directions and applications of NLP in healthcare documentation beyond comorbidities. How can this technology be further extended to benefit patient care and streamline medical records?

Further discussion of future directions has been added to the discussion on page 17. 

References:

Ensure that all references are up-to-date and accurately cited. Consider including recent works related to NLP applications in healthcare for a comprehensive literature review.

We have ensured that the references are up to date. 

7. PLOS authors have the option to publish the peer review history of their article (what does this mean?). If published, this will include your full peer review and any attached files.

Do you want your identity to be public for this peer review? For information about this choice, including consent withdrawal, please see our Privacy Policy.

Reviewer #2: No

Reviewer #3: No

Sincerely,

Aayush Setty

Rahul Sastry, MD

---

## [Decision Letter · Decision Letter 3]

5 Apr 2024

Natural Language Processing Augments Comorbidity Documentation in Neurosurgical Inpatient Admissions

PONE-D-22-22485R3

Dear Dr. Sastry,

We’re pleased to inform you that your manuscript has been judged scientifically suitable for publication and will be formally accepted for publication once it meets all outstanding technical requirements.

Kind regards,

Vijayalakshmi Kakulapati, Ph.D

Academic Editor

PLOS ONE

Additional Editor Comments (optional):

author addressed all comments  
